# Food Pleasure Profiles—An Exploratory Case Study of the Relation between Drivers of Food Pleasure and Lifestyle and Personality Traits in a Danish Consumer Segment

**DOI:** 10.3390/foods11050718

**Published:** 2022-02-28

**Authors:** Nikoline Bach Hyldelund, Derek Victor Byrne, Barbara Vad Andersen

**Affiliations:** 1Food Quality Perception and Society Team, iSense Lab, Department of Food Science, Faculty of Technical Sciences, Aarhus University, 8000 Aarhus, Denmark; derekv.byrne@food.au.dk (D.V.B.); barbarav.andersen@food.au.dk (B.V.A.); 2Sino-Danish Center for Education and Research (SDC), Beijing 101408, China

**Keywords:** food pleasure, food reward, food pleasure scale, profiling, segment, consumer science

## Abstract

A greater comprehension of factors contributing to pleasure from food-related experiences could increase understanding of underlying processes around different eating behaviours. We explored drivers of food pleasure and whether certain consumer characteristics were associated with specific food pleasure profiles. This study aimed to investigate (1) how Danish consumers vary in terms of primary drivers of food pleasure, and (2) how differences in food pleasure are related to specific sociodemographic, lifestyle, health and eating behavioural personality traits. Three-hundred and fifty-five respondents (mean age 33.3 years) rated the importance of different drivers of food pleasure, along with sociodemographic, lifestyle, health and eating behaviour variables. Segmentation analysis was performed based on emerging food pleasure dimensions, and profiling of segments was conducted by multivariate regression analysis and calculations of odds ratios. The results demonstrated that five specific consumer segments could be defined, ‘Sensory-pleasure Seekers’ (50%), ‘Internal-pleasure Seekers’ (34%), ‘Contextual-pleasure Seekers’ (17%), ‘Exploratory-pleasure seekers’ (13%) and ‘Confirming-pleasure seekers’ (5%), each with specific characteristics. Importantly, this research indicates that a link between mental health, personality, eating behaviour and perceived food pleasure is evident. These insights contribute to the comprehension of the complex nature of food choices of importance to accommodating public health issues.

## 1. Introduction

The reason why people eat from a mere physiological point of view is to sustain energy levels, avoid starvation and reach bodily homeostasis [1,2]. However, it is increasingly believed that food intake is primarily driven by hedonic factors, such as sensory cues, palatability and pleasure, and that these can overrule homeostatic-driven satiety signals and thereby lead to consumption above and below physiological needs [1,3,4,5,6]. A focus on the hedonic side of eating has been prevalent in food and consumer research in recent years as an attempt to better understand the complex daily decision-making of consumers with regards to food intake [7,8,9,10,11]. This has yielded a comprehensive, yet fragmented picture of the many factors that can affect the subjective pleasurable eating experience [6]. It is however not clear which of these many factors are driving the subjective experience of food pleasure in the individual consumer. Thus, it seems there is a need for a holistic view on food pleasure for truly comprehending the many different components that influence the unique and flexible food choices and eating behaviours of consumers.

### 1.1. Variables Affecting Food Pleasure

The scientific literature has given insights into a broad range of variables that can affect the pleasurable aspects of food and eating. First and foremost, the intrinsic product characteristics, such as the sensory properties of food, have been found to be highly related to both acceptance, degree of liking, food satisfaction and preference towards food items [12,13,14]. Thereby, the sensory properties have been determined to not only be an important factor for the sensory perception of a specific food but also to be able to evoke a hedonic component, which can motivate a person to eat even after satiation has occurred [6,7,12]. The pleasurable experience of eating a specific food can also be affected by expectations and prior experiences one may have with that particular type of food [14,15,16,17]. Deliza and MacFie developed a model for illustrating the effects of expectations on product selection and evaluation and hereby visualised how, if expectations are met, consumers will gain satisfaction and pursue the repeated purchase of that or similar products [17]. Conversely, if expectations are not met, the probability of repurchasing or reuse of that or a similar product is unlikely. Expectations can be formed from many different inputs from a food; especially the packaging and serving context will influence the generation of expectations [17,18,19]. Prior expectations based on fond memories and nostalgia related to a food can also create expectations that will lead to a higher degree of liking or satisfaction [6,13,15,16,20]. Moreover, being positively surprised by a food, of which expectations were low, has also been shown to enhance liking and satisfaction [14,21,22]. Extrinsic product characteristics, such as information on product origin, ethical values in relation to production methods, and nutritional information, have likewise been found to increase satisfaction and liking of food products [3,15,22,23]. Post-ingestive sensations of feeling satisfaction and a sense of physical or mental wellbeing after eating were also found important as components for consumers for food-related pleasure and satisfaction [8,15,24,25,26]. Lastly, the context a person dines in, both in terms of the physical environment in which the food is eaten, the atmosphere of the setting, as well as the social interactions during intake, have been found in numerous studies to have a high impact on amounts consumed, acceptance, satisfaction and liking of food [14,15,18,27,28].

Andersen et al. (2021) in a study on the hedonic side of eating, conceptually clarified how perceived food pleasure is affected by individual differences and preferences, but at the same time can form characteristic patterns across populations. The authors introduced a conceptual understanding of ’food pleasure’ as an umbrella term for hedonic function over a spectrum of different food-related behaviours, thereby uniting the many different factors that may affect individual perceived food-related pleasure. By understanding the individual drivers of food pleasure and their relative importance in a holistic integrative way, it becomes possible to understand the pleasurable side of eating and how that affects eating behaviour and food choices too. Furthermore, Andersen et al. introduced a conceptual framework for the development of a multi-dimensional measurement of food-related pleasure; the Food Pleasure Scale [6]. By having such an instrument, researchers will be able to accurately tap into the subjective nature of food-related pleasure as well as enable the identification of people with an altered or impaired hedonic response. By having these insights of food pleasure cues, our understanding of which conditions promote specific eating behaviours could be expanded as well. Thereby, it would be possible to clarify why some people tend to over- or under-eat while others do not, and guide these people towards a healthier and/or more sustainable eating behaviour.

### 1.2. Personalities, Lifestyles and Eating Behaviour

Meanwhile, just as there is a wide range of factors in relation to food and eating situations that are shown to affect food pleasure, there are also many variables associated with the individual consumer, which affect food choice and eating behaviour. Several studies have investigated the link between eating behaviour and personality traits, where what has been described as ‘the big five’ personality traits (extraversion/introversion, agreeableness, conscientiousness, neuroticism and openness) have been linked to various eating behaviour styles and food choices [29,30,31,32,33]. Keller and Siegrist (2015) found that personality traits directly and indirectly via eating styles influenced food choices [31]. Particularly, people with a neurotic personality type tended to consume more sweet and savoury foods, as a symptom of emotional and external eating behaviour. Conversely, they found that having a predominantly conscientious personality type would promote the adoption of regulatory dietary restraint [31]. Conscientiousness and openness have also been shown to be positively associated with a higher fruit and vegetable intake, as well as a lower meat intake [34]. Personality traits are regarded as stable. Thus, a person’s personality type may be a direct risk factor for an unbalanced diet, and the repercussions that may follow.

Different types of eating patterns related to motivation for eating have likewise been explored [3,19,35,36]. The ‘Externality theory’, which is the behaviour of eating in response to external cues rather than an internal sensation of hunger [37,38], the ‘Psychosomatic theory’, which is the behaviour of eating as a way of relieving anxiety [37] and the ‘Restraint eating theory’ (the theory that most people are either restraint or unrestraint eaters, and that the constant cognitive effort to resist the desire for eating in restraint eaters can, in the end, be such a burden that they will eventually overeat as to relieve strong emotions) [37,39,40], have all been developed and tested with the purpose of finding the reason for the development and maintenance of obesity [35,41]. In addition, several questionnaires, such as the Restraint Scale [42], the Dutch Eating Behaviour Questionnaire [41], and the Three-Factor Eating Questionnaire [36], have been tested and used in studies of eating behaviour, all focusing on restraint eating, emotional eating and uncontrolled eating. Thus, many different personality traits and eating behaviour styles have been found to affect the individual’s approach to food, eating and food reward.

Furthermore, the ability to perceive pleasure is believed to be associated with a person’s general health condition, and especially mental health [9,43,44,45]. Anhedonia, the lack of ability to perceive pleasure, is a common symptom of a wide range of diseases, including depression and schizophrenia [43,46,47]. Thereby, the ability of the individual to sense pleasurable experiences, including pleasurable food experiences, has been acknowledged as an essential part of human wellbeing and healthiness. Conditions of chronic and acute stress have also been linked to changes in eating behaviour. Studies have shown that prolonged periods of physical and mental overload can cause non-homeostatic hunger and overconsumption, as food intake can dampen the physiological and behavioural stress responses [48,49,50]. Especially, the hedonic experience of eating has been proposed to play a distinct part in the effect of stress-induced eating, as eating activates neural substrates, such as dopamine, in a similar manner to drug abuse [51,52,53,54,55,56]. Living in a stressful environment may alter what we find pleasurable, as well as how much food is needed to reach an optimal level of pleasure. The fact that people on a global scale are experiencing an increasingly stressful everyday life makes this issue especially earnest [57].

As illustrated in the text above, a wide number of consumer, food and sensory research studies have investigated the elements of food reward and how this is linked to both food choice and eating behaviour. These efforts have lead to a comprehensive amount of information on the subject, yet a very fragmented picture still exists on the core elements of the concept of pleasure of food. Thus, the research could benefit from having a more coherent understanding of the concept of food pleasure so as to more accurately be able to tap into what brings individuals pleasure from food and food-related experiences. Thereby, the foundation for better understanding human food choices and conditions that may promote specific eating behaviours could be laid. Furthermore, it is still unknown whether there are differences in terms of the importance of different drivers of food pleasure among normal healthy consumers, and if these possible differences can be explained by personality traits or eating behaviour characteristics, or on the contrary, if eating behaviours can be explained by an individual’s food pleasure profile. Greater insights into the relationship between individual drivers of perceived food pleasure and personality characteristics, lifestyle and health and eating behaviour patterns could thereby help answer questions such as; why some people tend to over-consummate whilst others do not, and why changing diet and intake patterns have proven one of the most difficult challenges for many modern consumers [58].

### 1.3. Aim

The overall purpose of this study was to investigate if certain consumer characteristics are associated with specific food pleasure profiles. Specifically, the study aimed to investigate:How Danish consumers vary in terms of primary drivers of food pleasure.How differences in food pleasure are related to specific sociodemographic, lifestyle, health and eating behavioural personality traits.

In relation to the first aim, it was hypothesised that general traits in perceived food pleasure can be found within different consumer segments, and thus different segments will have different primary drivers of food pleasure. Moreover, in relation to the second aim, it was hypothesised that different socio-demographic, lifestyles, health and eating behavioural personality traits are related to and may even affect the degree of importance ascribed to aspects of food pleasure, within specific food consumer segments. The study thereby concerned both qualitative and quantitative aspects of food pleasure, with a view to providing insights into what might influence food pleasure as well as individual food pleasure profiles.

## 2. Materials and Methods

### 2.1. Data Collection

A questionnaire was developed based on the items included in the original conceptual framework for the Food Pleasure Scale, as proposed by Andersen and colleagues, 2021 (co-authors on the present paper) [6]. The scale consisted of 21 different items, each representing an aspect previously found to affect the subjective pleasure from food and food-related experiences. These items were furthermore grouped according to common characteristics to form distinct dimensions of food pleasure, thus representing different areas that potentially could affect the subjectively perceived pleasure from food (e.g., the items ‘Appearance’, ‘Odour’, ‘Taste’ and ‘Texture’ were grouped into a dimension of sensory characteristics). Appendix A gives an overview of the items and dimensions as they were originally presented by Andersen et al. (2021). The process of developing the Food Pleasure Scale is an iterative process, which includes a series of studies on the topic. Therefore the methodology of this study builds upon insights of other previous explorative studies on the aspects of food pleasure, and for this reason, specific items (e.g., ‘Product information’ and ‘Ethical values’) were not included in this study. Moreover, in this study, one additional item, ‘Atmosphere’, was added to the questionnaire, as this has been shown to improve food acceptability and wellness associated with food [27,59,60]. For each item in the questionnaire, the respondents were asked to rate the importance of the specific item in relation to their general pleasurable experiences of food on a 5-point ordinal scale ranging from 1 (= ’Not important at all’) to 5 (= ’Extremely important’). Appendix A gives a full overview of the questions and response variables as phrased in the questionnaire. Besides the items based on the framework for the Food Pleasure Scale, the questionnaire included socio-demographic, lifestyle, health and eating behaviour-related variables. The socio-demographic variables included ‘gender’, ‘age’, ‘educational level’, ‘employment status’, ‘number of people in the residence’ and ‘having resident children’. The ‘lifestyle and health’-related variables included ‘diet type’, ‘level of physical activity’, ‘height’, ‘weight’, ‘smoking habits’, ‘alcohol consumption’, ‘perceived stress in everyday life’, ‘health worries’, the prior or current treatment of depression, diabetes, addictions and/or eating disorders and ‘own perception of personality type’ in relation to being an intro-/extrovert [61,62,63,64]. The ‘eating behaviour’-related variables investigated the general appetite of the respondents by ‘general level of appetite’, ‘general enjoyment of the taste of food’, as well as any ‘experienced changes in appetite’ due to illness. Furthermore, the ‘eating behaviour’ category sought to clarify the individual level of food neophobia via a Danish translation of the Food Neophobia Scale [65,66,67] and the eating behaviour type was likewise via a Danish translation of the Dutch Eating Behaviour Questionnaire [35,41]. A comprised version of the Dutch Eating Behaviour Questionnaire consisting of 16 questions (rather than 33 as in the original tool) was used to avoid exhaustion of the respondents [35]. The questionnaires were online from October 2020 to November 2020, and data was collected via the CompuSense^®^ Cloud software, CompuSense Inc., Version 21.0.7955.28103 (Guelph, ON, Canada) [68].

### 2.2. Recruitment

Three hundred and fifty-five (284 females; 71 males) Danish respondents with a mean age of 33.34 years (SD = 13.07) were recruited via social media with Facebook and LinkedIn as the main platforms, as well as the Danish research recruitment site (www.forsøgsperson.dk [Edit., ‘Eng: www.testsubject.dk’], accessed on 24 February 2022)). Approximately 280 respondents were pursued to ensure statistical power in the exploratory factor analysis. Ethical approval is not required for this type of study according to the National Committee on Health Research Ethics in Denmark (Section 14 (2) in the Committee Act) [69]. All respondents gave written consent to use their data prior to commencing on the questionnaire.

### 2.3. Statistical Analysis

All data analyses were executed in IBM© SPSS© Statistics, version 27 (Armonk, NY, USA) [70] and R Studio©, version 1.3.1093 (Boston, MA, USA) [71].

#### 2.3.1. Actual Scale Structure and Dimension-Specific Food Pleasure

To explore whether the proposed framework of the Food Pleasure Scale, including the composition of the twenty-one items grouped in five different dimensions, were truly reflecting the experience of food pleasure of the actual consumers, an exploratory factor analysis (EFA) using parallel analysis with a Varimax rotation was performed [72,73]. Hereby, underlying factors of the food pleasure variables as experienced by this sample were discovered, as well as redundant items were detected. In this study, the recommended methodology by Costello and Osborne, 2005, Hayton et al., 2004 and Stevens, 2009 for exploratory factor analysis was used. Factors with eigenvalues above one and above the corresponding eigenvalues of the parallel analysis were accepted. The exclusion criteria for items and factors were as such; items with an extraction value below 0.32 were rejected. Items that load on more than one factor as well as items that were not loading on a factor were considered redundant, and thus removed from the scale. Factors that had less than three items loading on them, were considered to be weak, and thus items loading on these factors were also removed as well as reducing the number of factors [72,73,74]. Only results of the final items were used in the consecutive analyses of the food pleasure variables. To evaluate the coherence of the items within each emerging factor, internal consistency was tested by calculation of Cronbach’s alpha scores as well as average inter-item correlations for each factor [74,75]. The Cronbach’s Alpha scores were interpreted as such; α ≥ 0.9 = Excellent, 0.7 ≤ α < 0.9 = Good, 0.6 ≤ α < 0.7 = Acceptable, 0.5 ≤ α < 0.6 = Poor, α < 0.5 = Unacceptable. The ideal range of average inter-item correlation was 0.15–0.50; a value below 0.15 indicated that the items were not well correlated. A value above 0.50 pointed to the items being so close as to be almost repetitive [76].

Calculations of descriptive statistics were made for all variables to obtain a complete overview of distributions and median values (Appendix A). To study the importance of each dimension of food pleasure, a dimension-specific food pleasure factor was calculated by averaging the scores of the importance of the items within that specific dimension. To determine a possible ranking of the dimensions, in terms of most important dimensions and items within food pleasure, a Kruskal-Wallis test was conducted for assessing statistical differences between the mean ratings of the dimensions, and a Dunn’s test was conducted to evaluate where potential differences lied. All statistical tests were carried out with α = 0.05.

#### 2.3.2. Segmentation and Profiling of Consumer Segments According to Most Important Driver for Food Pleasure

To further explore the characteristics of different food consumer groups in relation to food pleasure, a segmentation analysis was performed by use of the emerging food pleasure dimensions. Thus, the respondents were grouped in consumer segments by which of the dimensions they had rated highest. To examine the differences between the segments, Χ^2^ test values and corresponding *p*-values were calculated for all variables except for age, BMI, rating of food neophobia, rating of the three eating behaviour styles (restricted eater, emotional eater and external eater), and the dimension-specific food pleasure factor. Here Kruskal-Wallis tests were used. For profiling of the consumer segments, multivariate analysis by logistic regression and odds ratios were calculated for all variables including lower and upper levels for 95% confidence intervals. All calculations of odds ratios were adjusted for age and gender, as these two variables were believed to be possible confounders. All statistical tests were carried out with α = 0.05.

## 3. Results

### 3.1. Scale Structure and Items

The parallel analysis of the food pleasure variables resulted in a reduced 18-item scale with five factors accounting for a total of 57.2% of the variance of the data. The five factors explained; 23.3% (Factor 1), 11.3% (Factor 2), 8.5% (Factor 3), 7.6% (Factor 4), and 6.5% (Factor 5) of the variance, respectively. See Table 1 for an overview of the results of the exploratory factor analysis on the sample including loadings for each item.

Based on the criteria set out for item and factor reduction, no items were removed. The five factors were tested for internal reliability to evaluate the coherence and accuracy of the factors. The Cronbach’s alpha scores for the factors were 0.74 (Factor 1), 0.68 (Factor 2), 0.72 (Factor 3), 0.59 (Factor 4) and 0.63 (Factor 5), which were all good or acceptable alpha scores. Factor 4 was retested for internal reliability, as the alpha score for this factor was just below that being acceptable. Thereby the item ‘Choice’ was removed, as this item had the lowest extraction value within the factor, and the result after exclusion of ‘Choice’ was an alpha score of 0.62. This item was thereby removed to improve the strength of this factor. The total reliability of the 18-item scale was defined as good (α = 0.76). Average Inter-Item correlations were also calculated for each factor and for the total scale. The average inter-item correlations were 0.37 (Factor 1), 0.35 (Factor 2), 0.46 (Factor 3), 0.35 (Factor 4), 0.36 (Factor 5) and 0.16 (total scale), all within the optimal range. In order to preserve the original terminology of the Food Pleasure Scale, the ‘factors’ will be referred to as ‘dimensions’ from this point on.

### 3.2. Drivers of Food Pleasure

The median values and interquartile range for ratings of the importance of every item can be seen in Appendix A. The first dimension consisted of the items; ‘Food texture’, ‘Food odour’, ‘Food appearance’, ‘Food taste’ and ‘Pleased senses’ (Table 1). These are all items that represent the perception of food pleasure from a sensory perspective, and thus this dimension was named the ‘Sensory-driven pleasure’ dimension. The second dimension included the items; ‘Novelty’, ‘Variation’, ‘Surprise’ and ‘Memory’. These are items that represent an exploratory perspective of food pleasure, and the dimension was thereby named the ‘Exploratory-driven pleasure’ dimension (Table 1). The third dimension consisted of ‘Atmosphere’, ‘Physical surroundings’ and ‘Eating with others’. These items reflect a side of food pleasure that has to do with the context of the eating experience, and therefore the dimension was named the ‘Contextual-driven pleasure’ dimension (Table 1). Dimension four, which consisted of ‘Habit’, ‘Eating alone’ and ‘Familiarity’, was named the ‘Confirming-driven pleasure’ dimension, as these items represent a perspective of food pleasure that has to do with comfort and confirmation within a meal (Table 1). Lastly, the fifth dimension consisted of the items; ‘Mental sensation’, ‘Physical sensation’ and ‘Need’. These items together represent a perspective of food pleasure that revolves around the interoceptive sensations and inner experience of eating. This dimension was therefore named the ‘Internal-driven pleasure’ dimension (Table 1). Hence, the drivers of food pleasure in this sample were characterised by a main focus on the sensory attributes, with four secondary dimensions representing an explorative, contextual, confirming or internal approach to the experience of food pleasure. A Kruskal-Wallis ANOVA showed that the five dimensions were rated differently from each other in terms of the level of importance (*p* < 0.001), and following a Dunn’s test for pairwise multiple comparisons, it was found that this difference lay between all the dimensions (*p* < 0.001 for all pairwise comparisons), except between the ‘Contextual-driven pleasure’ and ‘Exploratory-driven pleasure’ dimensions (*p* = 0.412). Thereby, the ‘Sensory-driven pleasure’ dimension was rated most important for the complete sample of respondents with a mean rating of 4.12 (±0.59), followed by the ‘Internal-driven pleasure’ dimension, with a mean rating of 3.86 (±0.71). The ‘Contextual-driven pleasure’ and ‘Exploratory-driven pleasure’ dimensions shared the position of being third most important, with similar mean ratings of 3.53 (±0.79) and 3.56 (±0.72), respectively. Finally, the ‘Confirming-driven pleasure’ dimension was rated least important, as the mean rating was 2.62 (±0.82).

### 3.3. Profiling of Consumer Segments Defined by Most Important Food Pleasure Dimension

Five segments were defined based on the pleasure dimension with the highest relative importance rating of each respondent. A sub-group of respondents, who had a tie of two or more dimensions as most important, were also included in the segmentation analysis. The segments were named as such; The ‘Sensory Pleasure Seekers’ (n = 176), the ‘Internal Pleasure Seekers’ (n = 121), the ‘Contextual Pleasure Seekers’ (n = 60), the ‘Exploratory Pleasure Seekers’ (n = 46) and the ‘Confirming Pleasure Seekers’ (n = 18). The ‘Tie on first’ subgroup consisted of n = 46 respondents, and thus was larger than the ‘Confirming Pleasure Seekers’ in terms of count of members. An overview of the five segments including the ‘Tie-on-first’ group and the main characteristics of each segment can be seen in Figure 1. For a complete overview of distributions of all variables by the segments as well as results of Χ^2^/Kruskal-Wallis tests, see Appendix A. In the following paragraphs, in-depth profiling of each segment will be described in relation to the socio-demographic, lifestyle, health and eating behaviour variables. For a complete overview of the results of the multivariate regression analysis by odds ratios see Appendix A.

#### 3.3.1. The ‘Sensory Pleasure Seekers’

The ‘Sensory Pleasure Seekers’ were characterised by being primarily driven by the sensory-driven pleasure dimension with a very high likelihood of 2260% of being rated most important. The sensory-driven pleasure dimension was characterised by the items: ‘Food appearance’, ‘Food odour’, ‘Food taste’, ‘Food texture’ and ‘Pleased senses’ (Table 1). On the other hand, the segment had a decreased likelihood of 59% of rating the internal-driven pleasure dimension high. Furthermore, this segment had increasingly high likelihood of having passed an education above high school level. Odds ratios were especially high for having a vocational education (OR. 7.00), short higher education (OR: 5.12) or a medium higher education (OR: 4.51). At the same time, this segment had a decreased likelihood of 77% of being students compared to being an employee. Furthermore, belonging to this segment increased the likelihood of having two children in the residency. Moreover, this segment had a decreased likelihood of 60% for often feeling stressed compared to sometimes feeling stressed. At the same time, they had an increased likelihood of 173% of having received treatment for stress prior to participating in this study. No other characteristics in relation to lifestyle, health, appetite and eating behaviour could be determined.

#### 3.3.2. The ‘Internal Pleasure Seekers’

The ‘Internal Pleasure Seekers’ had increased likelihoods of regarding internal-driven pleasure (OR: 27.63) and confirming-driven pleasure (52%) as most important. The internal-driven pleasure dimension was characterised by the items: ‘Mental sensation’, ‘Physical sensation’ and ‘Need’ (Table 1). Oppositely, there were decreased likelihoods of rating the three other pleasure dimensions high, with sensory-driven pleasure being least likely of a high importance rating. This segment showed, as the only segment, to have an increased likelihood of 171% of being male. Furthermore, the members of this segment had significantly high odds ratios for being self-employed (OR: 19.36) as compared to being an employee. The segment also had an increased likelihood of 36% of having a predominantly introvert personality type, and oppositely, a decreased likelihood of 22% of having a predominantly extrovert personality type. Moreover, a decreased likelihood of currently receiving diabetes treatment of 73% could be detected. The ‘Internal Pleasure Seekers’ also proved to have a significantly large (OR: 2.14) or very large (OR: 3.23) general appetite. They were furthermore characterised as both emotional eaters (OR: 1.37) and external eaters (OR: 1.64).

#### 3.3.3. The ‘Contextual Pleasure Seekers’

The ‘Contextual Pleasure Seekers’ were characterised by an increased likelihood of 1927% of being driven by the contextual-driven pleasure dimension. The contextual-driven pleasure dimension was characterised by the items: ‘Atmosphere’, ‘Physical surroundings’ and ‘Eating w. others’ (Table 1). On the contrary, they had a decreased likelihood of 54% and 39% of rating the sensory-driven and internal-driven pleasure dimensions high, respectively. In addition, this segments was characterised as being students (OR: 2.73), and likewise they had decreased likelihoods of having finished a vocational education (80%), a medium higher education (79%) and a long higher education (62%) compared to a high school degree. This segment was typically only one person in the household (OR: 2.73), as well as they had an increased likelihood of 152% of having one child in the residency. The ‘Contextual Pleasure Seekers’ showed no specific lifestyle, health, appetite or eating behaviour characteristics.

#### 3.3.4. The ‘Exploratory Pleasure Seekers’

Members of the ‘Exploratory Pleasure Seekers’ segment was characterised by having an increased likelihood of 1494% of rating the exploratory-driven pleasure high. The exploratory-driven pleasure dimension consisted of the items: ‘Surprise’, ‘Novelty’, ‘Variation’ and ‘Memory’ (Table 1). In addition, the segment had a decreased likelihood of 58% and 48% of rating the sensory-driven pleasure dimension and the internal-driven pleasure dimension high on the importance scale, respectively. Moreover, the segment had an increased likelihood of 331% of being vegetarian, as well as 241% of being prior smokers. This segment was also characterised by a decreased likelihood by 31% of having a predominantly introverted personality. Furthermore, the ‘Exploratory Pleasure Seekers’ proved as the only segment to have an increased likelihood of currently, as well as previously, having received treatment for diabetes of 468% and 665%, respectively. Within the total study population, 63 (18%) respondents were receiving treatment for diabetes at the time they answered the questionnaire, and 38 (11%) respondents had received diabetes treatment prior to the investigation. Within the ‘Exploratory Pleasure Seekers’, 10 (22%) respondents were currently in treatment for diabetes, and 7 (15%) had previously been in treatment for diabetes, respectively (Appendix A). The segment showed no specific characteristics in relation to sociodemographic variables, appetite or eating behaviour.

#### 3.3.5. The ‘Confirming Pleasure Seekers’

The ‘Confirming Pleasure Seekers’ were primarily driven by confirming-driven pleasure (OR: 67.33). The confirming-driven pleasure dimension consisted of the items: ‘Habit’, ‘Familiarity’ and ‘Eating alone’ (Table 1). On the contrary, this segment was not driven by sensory- or exploratory-driven pleasure, with odds ratios of 0.14 and 0.29 respectively. This segment showed no specific characteristics in terms of socio-demographic, lifestyle and health variables, with one exception; they had an increased likelihood of 583% of having a predominantly introvert personality type. This segment did however show tendencies towards having increased likelihoods for previously having received treatment for an eating disorder with an increased likelihood of 1675%. Within the total study population, 22 respondents (6%) had previously been treated for an eating disorder. Three of these respondents were part of the ‘Confirming Pleasure Seekers’ segment, which corresponded to 17% of the respondents within this group (Appendix A). Furthermore, this segment was characterised as the only segment that a decreased likelihood of 18% of being food neophillic, as well as an increased likelihood of 344% of being restricted eaters.

#### 3.3.6. The ‘Tie-On-First’ Subgroup

The ‘Tie-on-first’ subgroup had an increased likelihood of 134% of rating the contextual-driven pleasure dimension high. The group was not characterised by any socio-demographic variables. However, this group showed some interesting characteristics in terms of lifestyle and health. The group proved to have a significantly increased likelihood of 347% of feeling depressed in general as well as an increased likelihood of previously having received treatment for an eating disorder of 329%. At the same time, they had a decreased likelihood of 63% of rarely feeling stressed compared to sometimes feeling stressed.

## 4. Discussion

### 4.1. Food Pleasure as a Multi-Facetted Concept

Results of the complete sample reflected that food pleasure was in general perceived to be important, as the mean importance ratings of the five dimensions and median ratings of the 18 items all were rated above the centre point of the 5-point rating scale. The ‘Sensory-driven pleasure’ dimension, and its appertaining items, were rated as the most important aspects for the experience of food pleasure by the total sample. This result confirms previous findings of the sensory aspect of food being closely related to food hedonia and food-related wellbeing [12,15,24], as well as acceptance, liking and food choice and behaviour [13,14,77]. In addition, we found that having an experience of one’s senses being positively stimulated in a collected way was regarded as equally important as the individual sensory properties to the experience of pleasure. The ‘Internal-driven pleasure’ dimension and its associated elements were rated second highest by the sample, which indicated that internal bodily sensations in relation to food consumption have a noteworthy effect on pleasure perceived from food. Previous studies have found similar results, where especially the post-ingestive physical and mental sensations were found to be of importance to food satisfaction and joy [10,15,24,25,45,78]. Exploratory-driven and contextual-driven pleasure were found to be equally important to the respondents. The contextual-driven pleasure aspects are all elements that add to the experience of pleasure by surrounding the consumer and creating a specific environment. The context of eating has been widely described as one of the variables that can affect the liking and acceptability of food [14,18,77,79]. Likewise, the respondents also regarded the ‘Exploratory-driven pleasure’ dimension to be important, fitting largely with research describing how psychological factors such as arousal, previous memories and personality traits have effects on food choice and behaviour [14,21,80,81,82]. Thus, it seems plausible that this pleasure dimension was manifest in this sample too. Finally, the least important dimension for this sample as a collected group was the ‘Confirming-driven pleasure’ dimension. The items of this dimension; ‘Familiarity’, ‘Habit’ and ‘Eating alone’, have been found to be related to emotional eating in terms of being aspects of food pleasure that can induce calmness and relieve from stress [50,64,83,84]. The fact this dimension was evident from the exploratory factor analysis shows this is an aspect of food pleasure that has importance to some consumers, though not the majority. The item of ‘Eating alone’ had a median rating of 2.00 (1.5–3), thus indicating that this aspect was not important for food pleasure for this sample. Nonetheless, this aspect may still be important to some specific consumer segments, e.g., elderly people suffering from dysphagia as reported in Tibbling and Gustafsson (1991) [85]. No people above the age of 69 years were represented in this study, which may in part explain why this specific item was not found to be important for food pleasure.

### 4.2. Consumer Segments Defined by Most Important Drivers to Food Pleasure

This study has shown that drivers of food pleasure can be a useful tool for defining consumer segments, as distinct characteristics of the segments were detected. A total of 50% of the respondents were characterised as ‘Sensory Pleasure Seekers’, meaning that the sensory characteristics of food were the primary reason for food-related pleasure, for the majority of consumers. This finding emphasises once again the importance of the sensory aspects of food to the perception of food pleasure. This segment was characterised by having an education above high school level as well as not being a student and having two children in the household. As this segment represents half of the study sample, these characteristics may simply be an expression of what the majority of consumers look like; regular working families, with a regular appetite and preferences in terms of food pleasure towards the intrinsic product characteristics of the food. Living a somewhat stressful everyday life with children, which may also not leave much room for reflection on food preferences, nor time for a focus recognising internal wellbeing from food-related experiences [49,50,86].

The ‘Internal Pleasure Seekers’, who were characterised by regarding internal pleasure sensations as most important, made up 34% of the respondents. This segment was characterised by being self-employed and having a predominantly introverted personality type. They also had a large appetite and showed no difficulties with feeling hunger. Furthermore, they were characterised as both emotional and external eaters, which means that this segment is likely to be motivated to eat as a reaction to negative emotions or external cues, rather than appetite sensations. This appears to highlight a segment, where eating indeed is governed by one’s internal state and emotions. Previous studies have found that emotional and external eaters are often characterised by not having an extrovert or conscientious personality type, but on the contrary have a higher tendency to be introverted, experience higher levels of anxiety or show traits of neuroticism [30,31,32]. Moreover, the segment also showed a decreased likelihood of having received treatment for diabetes. Diabetes has been characterised as one of the largest disease burdens of modern times and continues to increase in numbers in almost all countries [87,88]. The most common risk factors for type 2 diabetes are often ascribed to unhealthy diet, obesity and inactivity. Clinical variables, such as depression, and specific personality traits have also been associated with a higher risk of developing diabetes [87,88,89]. More specifically, a meta-analysis of five cohort studies found an elevated risk of diabetes for people with low levels of conscientiousness traits [89], where the underlying mechanisms of this finding were likely to be poor weight management and physical inactivity [33,88,89]. The decreased likelihood of having received diabetes treatment in the ‘Internal Pleasure Seekers’ segments may thereby be an expression of this segment being introverts, who are known to have high levels of conscientiousness [31], as well as experts in evaluating interoceptive sensations. On the other hand, whether this segment is capable of reacting in a sensible manner to these sensations is not clear, as they also have increased likelihoods of having an emotional or external eating behaviour, which has been shown to increase intake of highly palatable and energy-dense foods [30,31,32]. Thus, the findings point at the relevance of studying in the future, the ability of highly interoceptive people and internal pleasure seekers in making healthy food choices.

The ‘Contextual Pleasure Seekers’ were characterised by being students, and, in line with this result, not having finished a vocational, medium higher or long higher education. They are typically one person in the household, and they have a tendency to be extroverts. This segment showed no other characteristics in relation to lifestyle, health, appetite or eating behaviour. However, their sense of food pleasure was naturally driven by the contextual-driven pleasure dimension, which included the items ‘Atmosphere’, ‘Physical surroundings’ and ‘Eating w. others’, and oppositely not by the sensory-driven or internal-driven pleasure dimensions. In previous studies, it has been found that the meal context is correlated with higher intake levels as well as higher acceptability and liking of foods [5,18,28,60,79,90]. Wansink (2004) differentiates between two different types of contextual environments which affect food consumption levels; the ‘eating environment’ and the ‘food environment’, where the first concerns the atmosphere and sociability of the meal, and the latter regards the physical state of the food such as the shape, serving size and packaging [91]. Perhaps this segment represents a group of consumers that are most concerned with the eating environment and social aspects of eating, as they are characterised as extrovert students.

The ‘Exploratory Pleasure Seekers’ showed significant characteristics in relation to lifestyle and eating behaviour only. They appear to be vegetarians and prior smokers, as well as not being introverts. Opposite to introvert persons, extrovert persons have been found to have a larger intake of vegetables and fruits, as well as sweets, snack foods, meat and soft drinks [31]. This may in part explain the higher likelihood of being vegetarians in this segment. Furthermore, they appear as the only segment to have an increase in the prevalence of diabetes. These results seem somewhat contradictory, as a higher intake of vegetables and fruit has been shown to decrease the prevalence of diabetes [89,92,93,94]. On the other hand, the risk of diabetes has been found to be positively correlated with high extraversion and low conscientiousness personality traits [88,89]. Approximately 22% of the respondents in the ’Exploratory Pleasure Seekers’ segment replied they were receiving treatment for diabetes at the time of answering the questionnaire. However, when looking closer into the results of the different segments, the ’Confirming Pleasure Seekers’ do actually have a larger proportion of respondents who testify to having received treatment of diabetes, both before and at the time of answering the questionnaire. The ’Confirming Pleasure Seekers’ had six respondents (corresponding to 33% of the group) who reported being currently treated for diabetes, and five (corresponding to 28% of the group) who previously had treatment for diabetes. However, as this group is relatively small compared to the other segments (n = 18), calculations of odds ratios were not fruitful for these specific variables, and thereby these results were not reflected by the multivariate analysis. Nevertheless, the relation between specific diseases including diabetes and pleasure profiles needs further exploration. The ‘Exploratory Pleasure Seekers’ were furthermore characterised by preferring the exploratory-driven pleasure dimension, which means that this segment’s sense of food pleasure was driven by collative aspects of pleasure such as novelty, surprises and variation. These results match those observed in previous studies. Andersen and Hyldig (2015) found that variation, novelty and positive surprise was important factors for achieving a sensation of satisfaction after eating [10]. Likewise, Berlyne (1970) found that the optimal arousal level to be obtained from food and food-related experiences is influenced by a food product’s collative qualities [82]. Oppositely, a decreased preference for the sensory-driven and internal-driven pleasure dimensions was detected in this segment, thereby underpinning that food pleasure for this segment is not driven by the food itself or interoceptive sensations, but perhaps more by aspects relating to the complete experience of eating the food. Even though the information about this segment is scarce, the results does paint a picture of a segment that is seeking sensations in relation to both different diet types and stimuli.

Finally the smallest segment, the ‘Confirming Pleasure Seekers’, was characterised by being predominantly an introvert, as well as showing tendencies towards previously having received treatment for an eating disorder. This finding agrees with current literature, as it has been found that neuroticism traits are linked to depression, anxiety and eating disorders as well as an introvert personality [31,32,65,95]. Furthermore, this segment had a decreased likelihood of being food neophilic, as well as a higher prevalence of being restricted eaters. These results confirm previous findings of people with food neophobic traits having more restricted eating behaviours, as well as seeking food that will comfort and relieve mental stress [62,65,86,96]. This segment naturally preferred the confirming-driven aspects of food pleasure, and oppositely did not prefer the sensory-driven nor the exploratory-driven pleasure aspects. This shows that there is an evident link between mental health, eating behaviour and importance to aspects of food pleasure, and feelings of distress may be sought to be accommodated by eating comforting food.

The ‘Tie-on-first’-subgroup (n = 46) was larger in numbers than the ‘Confirming Pleasure Seekers’ (n = 18) and even in size with the ‘Exploratory Pleasure Seekers’ (n = 46). This gives indications that to the individual consumer, food pleasure could also be perceived as a multi-faceted construct, which could be driven by a range of different aspects. In addition, some consumers may experience having a mix of food pleasure preferences too. The subgroup also proved to have some specific characteristics, as this group had an increased likelihood of both feeling depressed as well as having received treatment for an eating disorder. These results do not reveal which eating disorders the respondents have been in treatment for. However, studies have found that the reward mechanisms involved in eating disorders are different from each other [43,83], with especially food addiction and binge-eating disorder being linked to an impaired sense of reward [4,43,97]. To the authors’ knowledge, no studies have investigated the link between food pleasure and having a presumably impaired reward system. Future studies on the topic are therefore recommended.

### 4.3. Strengths and Limitations of the Current Study

This study was envisioned as an exploratory to contribute to the further scientific examination of perceived pleasure from food, herein the differences in drivers of food pleasure and characteristics of different food pleasure profiles across segments of consumers—in this context within a Danish sample. The aims of the study were successfully met by the use of the elements of the proposed framework for the Food Pleasure Scale [6] as the backbone of a segmentation analysis. The exploratory factor analysis gave insights into the underlying dimensions of food pleasure as perceived by this sample, and paved the way for the segmentation analysis. The reduction of the items of the scale, and the resulting reliability tests confirmed the results of the exploratory factor analysis, as the scale, as well as each dimension of the scale, proved to have good or acceptable internal consistency.

As many of the characteristics of the food pleasure profiles discovered in this study are confirmed by previous studies using a range of different scales for attaining information on food choice and eating behaviour, it could be argued that the development of a Food Pleasure Scale could replace the use of these scales, as it provides much of the same information from using these alone. Similar results from scales such as the Food Neophobia Scale and the Dutch Eating Behaviour Questionnaire could possibly be attained from the Food Pleasure Scale too. The conceptual construction of the Food Pleasure Scale was indeed based on a comprehensive review of previous research on the hedonic side of eating, as well as an understanding of the concept of food pleasure as a holistic and all-encompassing term [6]. Thereby, it makes sense that the results of using the Food Pleasure Scale would reflect a broad and holistic approach to the hedonic motivation for consumption too, including aspects such as food neophobia and eating behaviour personality traits. By having a scale, which incorporates all of these aspects, researchers will be able to more precisely make use of insights into what brings individuals pleasure from food. A better understanding of the unique and flexible process of human food choices and conditions that may promote specific eating behaviours will thereby be possible from a single questionnaire. As the current study was of an exploratory nature, studies of larger sample sizes, including different consumer groups, are therefore needed to confirm this hypothesis.

Although the study was conducted on a sample of Danish consumers, the sample cannot be considered representative of the general Danish population. This would require a larger sample size and representation of consumer segments as found in the Danish population. However, due to the exploratory nature of this study, the study can be used to point researchers in the direction of segments with anhedonic traits. Especially consumers suffering from mental health issues, such as eating disorders, depression and stress are relevant for future research, as both eating disorders and mental illnesses are linked to impaired reward systems and the experience of anhedonia [43,52,98,99]. Furthermore, it seems relevant to investigate the food pleasure profiles of diabetics, as this study indicates a possible link between the importance of drivers of food pleasure and diabetes treatment. Similarly, previous research has proposed a relationship between pleasure and BMI status [11,100]. Yet, it is not clear if the sense of food pleasure, in general, is suffering across the different drivers of food pleasure, or whether dimension-specific anhedonia can occur. On that note, it would also be relevant to investigate whether, in the case of dimension-specific anhedonia, another food pleasure dimension could take over to compensate for the general perception of food pleasure. Recent research on patients suffering from sensory impairments due to COVID points at such possible adaption behaviour [101]. Further insights into this specific issue could assist health professionals in their work on treating people experiencing ageusia, anosmia and loss of appetite sensations, such as people with an eating disorder, the elderly and COVID-patients with sequelae.

Further, as food pleasure is regarded as a dynamic construct, drivers of food pleasure are also expected to vary over time dependent on the consumers and present state of life and mind. To further test the applicability and accuracy of the scale, studies examining the capability of the scale in measuring dynamics over time would be relevant. Validation of the performance of the Food Pleasure Scale in studies with people with presumed impaired reward systems, by use of different means, such as comparing results with neuroimaging or results from the Snaith-Hamilton Pleasure Scale [102], is also recommended to confirm construct validity of the scale.

## 5. Conclusions

This study set out to investigate if certain consumer characteristics are associated with primary drivers of food pleasure. It was discovered that specific consumer characteristics were indeed associated with specific food pleasure profiles, and the pleasure consumers perceive from food can be described both quantitatively and qualitatively by use of the elements of the framework for the Food Pleasure Scale. Thereby, drivers of food pleasure were also detected and it was found that sensory-driven, intrinsic-driven, exploratory-driven, contextual-driven and comforting-driven pleasure dimensions drove this sample of Danish consumers.

One of the more significant findings of this study was that the applicability of the elements of the framework for the Food Pleasure Scale in consumer segmentation studies proved a success. Five distinct consumer segments, each with their own specific characteristics, were detected. A segmentation analysis by multivariate regression analysis and odds ratio calculations showed that the study population could roughly be divided into one main, one secondary and three minor segments. The main segment was named the ‘Sensory Pleasure Seekers’ (n = 176, 50%) and their food-related pleasure was primarily driven by sensory aspects. Furthermore, this segment was characterised by having an education above high school level, not being a student and having two children in the household. The second-largest segment, the ‘Internal Pleasure Seekers’ (n = 121, 34%), was characterised as consumers whose pleasure was governed by their internal state and emotions in relation to personality, appetite and eating behaviour. The ‘Contextual Pleasure Seekers’ (n = 60, 17%) were characterised as extrovert students, whose, food pleasure was driven by social and contextual aspects, and oppositely not by sensory and internal aspects. The information retrieved about the ‘Exploratory Pleasure Seekers’ (n = 46, 13%) was scarce, however, the results painted a picture of a segment that is seeking sensations both in relation to different diet types and stimuli to experience food pleasure. Lastly, the ‘Confirming Pleasure Seekers’ (n = 18, 5%) was characterised by being predominantly introverted, and previously having received treatment for an eating disorder. This segment bared witness of a link between mental health, eating behaviour and perceived food pleasure. Further investigation of these specific correlations is suggested for future studies; as such, insights will contribute to the further comprehension of the complex nature of food choices and eating behaviour and could have significance in regards to accommodating public health issues.

## Figures and Tables

**Figure 1 foods-11-00718-f001:**
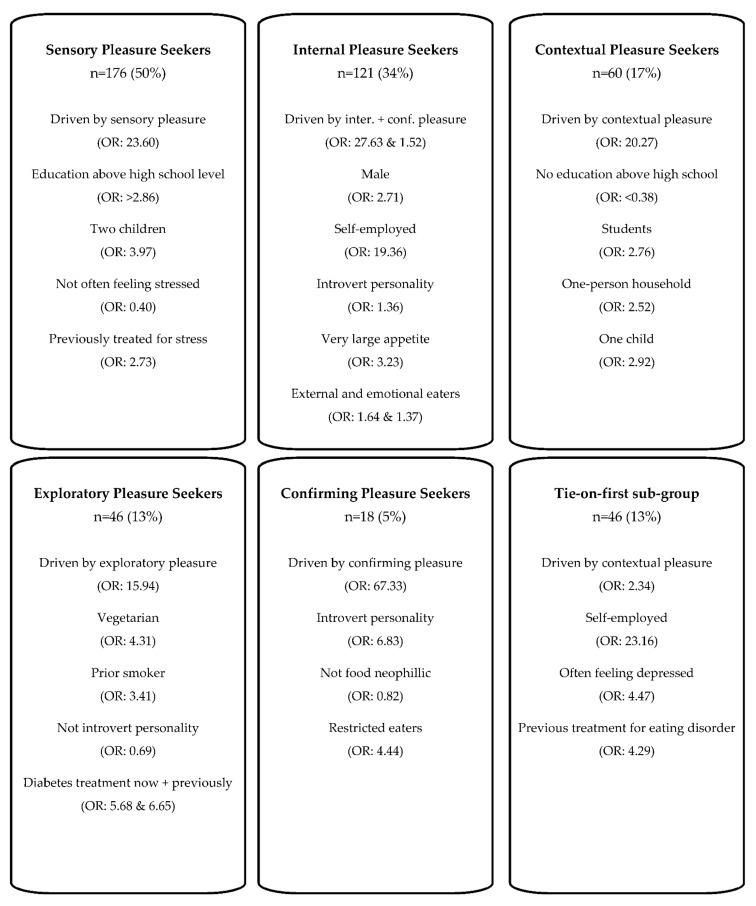
Overview of food pleasure segments including main characteristics of profiling by multiple logistic regression analysis and calculation of odds ratios (OR).

**Table 1 foods-11-00718-t001:** Results of exploratory factor analysis showing the rotated component matrix with item loadings on the five factors.

Items	Factors (Explained Variance of Data)				
	Factor 1 (23.3%)	Factor 2 (11.3%)	Factor 3 (8.5%)	Factor 4 (7.6%)	Factor 5 (6.5%)
Food texture	0.74				
Food odour	0.69				
Food appearance	0.67				
Food taste	0.66				
Pleased senses	0.59				
Novelty		0.83			
Variation		0.71			
Surprise		0.53			
Memory		0.46			
Atmosphere			0.79		
Physical surroundings			0.76		
Eating w. Others			0.69		
Habit				0.78	
Eating alone				0.70	
Familiarity				0.65	
Mental sensation					0.83
Physical sensation					0.82
Need					0.53

Extraction Method: Principal Component Analysis. Rotation Method: Varimax with Kaiser Normalization.

## Data Availability

The datasets generated for this study are available on request to the corresponding author.

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
