# Peer review of "Food Pleasure Profiles—An Exploratory Case Study of the Relation between Drivers of Food Pleasure and Lifestyle and Personality Traits in a Danish Consumer Segment"

_foods, 2022, doi:10.3390/foods11050718_

Round 1

Reviewer 1 Report

The aim of this study was to study Danish consumer sample on drivers of food pleasure and related factors. The study was executed as a survey for 355 consumers, and a segmentation analysis was performed based on food pleasure scale answers. Technically the study was performed according to good principles and review of the literature was sufficiently focused. In general, the manuscript is well structured and is easy to read.

However, the study has weaknesses in the consumer sample and in the survey content, which lead to a lack of novelty and impact of the results. This type of segmentation study would benefit clearly if the respondent sample is either country representative or then clearly specific target group.

Secondly, as the author states in the introduction there are many different variables affecting food pleasure, but in the scale used the items measured are very similar to each other and produce rather obvious result. It would have been much interesting if all the the key aspects from the food-related pleasure (Fig S1) would have been involved into the questionnaire dimensions measured (Table S1). Eg. eating healthy food or eating ethically produced food can produce pleasure experiences, which can even have different inportance.

Also, a question arises, that a scale, where all 18 items showed median ratings above the centre point of the 5-point rating scale, is a very good measuring scale. When the variation in the scale usage is very small, are all small nuances even meaningful results or just result from the random scale usage of the participants. The result that food sensory experiences (food texture, food taste, appearance and odour) are important for consumers is clearly shown already after Food Choice Queastionnaire studies since 1995.

The study could have been offered some new insights if the study would have involved some product tasting or ie. the segmentation could have been connected to evaluation/purchase intent of concept products.

Most the relevant tables were attached as supplementary material. It would make the article more easy to read if the main materials would be part of the main text.

Reviewer 2 Report

Dear authors,

The study is very interesting and well conducted. The manuscript presents a good quality and well described and discussed topic.

I have some few comments to improve the manuscript.

In the methods section, were The life-style and health variables  grouped from different questionnaires? or was it developed by the researchers for this study?

For eating behavior, authors mention the use of the Food neophobia scale and the Dutch one. Were they modified, or used as the original scale?

Was The Food neophobia scale applied in English? Was it validated for other languages?

Was The comprised version of the Dutch Eating previously validated?

Results

Table 1 is not necessary, just give the data inside the text.

Table S1 is not mentioned inside the manuscript, starting with S2, but S1 exists in the supplementary file.

Besides that, I do not have more questions.
